# An optimized approach to study nanoscale sarcomere structure utilizing super-resolution microscopy with nanobodies

**Collin M. Douglas**[1], **Jonathan E. Bird**[2☯], **Daniel Kopinke**[2☯], **Karyn A. Esser**[1]*

**1** Department of Physiology and Aging, University of Florida, Gainesville, Florida, United States of America,
**2** Department of Pharmacology and Therapeutics, University of Florida, Gainesville, Florida, United States of America

☯ These authors contributed equally to this work.
* kaesser@ufl.edu

**Data Availability Statement:** All data are in the manuscript and/or Supporting information files.

## Abstract

The sarcomere is the fundamental contractile unit in skeletal muscle, and the regularity of its structure is critical for function. Emerging data demonstrates that nanoscale changes to the regularity of sarcomere structure can affect the overall function of the protein dense ~2μm sarcomere. Further, sarcomere structure is implicated in many clinical conditions of muscle weakness. However, our understanding of how sarcomere structure changes in disease, especially at the nanoscale, has been limited in part due to the inability to robustly detect and measure at sub-sarcomere resolution. We optimized several methodological steps and developed a robust pipeline to analyze sarcomere structure using structured illumination super-resolution microscopy in conjunction with commercially-available and fluorescently-conjugated Variable Heavy-Chain only fragment secondary antibodies (nanobodies), and achieved a significant increase in resolution of z-disc width (353nm vs. 62nm) compared to confocal microscopy. The combination of these methods provides a unique approach to probe sarcomere protein localization at the nanoscale and may prove advantageous for analysis of other cellular structures.

## Introduction

Histology has historically been used to investigate the structure of biological tissues to better understand their function [1]. The sarcomere—the individual functional unit of skeletal muscle—offers a prime example of a structure-function relationship in biological tissues [2–4]. Sarcomeres are defined by their protein dense boundaries, known as Z-discs, and are otherwise largely composed of, actin, myosin, and titin proteins [5–9]. Numerous studies demonstrate that changes in the organization of these three major filamentous proteins affect the primary function of the sarcomere—to generate tension [6,10–26]. Further, although microscopic in size (~2.3–2.47 μm in total length), sarcomeres contain numerous additional proteins that localize to distinct domains that are important for its function [27]. These proteins localize in a region-specific manner, and failure to maintain these specific nanoscale localizations changes

**Funding:** This work was supported by National Institutes of Health grant 5R01AR079220 to Karyn A Esser and National Institutes of Health grant 1R01AR079449 to Daniel Kopinke. The funders had no role in the study design, data collection and analysis, decision to publish, or preparation of the manuscript.

**Competing interests:** The authors have declared that no competing interests exist.

functional capacity [12,16,17,19,28–33]. Thus, it can be appreciated that sarcomere function is dependent on both the amount and localization of its many proteins.

The mechanisms behind skeletal muscle weakness across numerous diseases are not fully understood [34]. Attempts to understand changes in skeletal muscle function with disease often focus on cross-sectional area, due to the common observation of significant muscle atrophy [20,22,35–46]. However, changes in muscle function often precede changes in muscle mass, and muscle mass does not positively correlate with the quality of sarcomere structure [37–39,42,43,47]. Additionally, while numerous chronic or genetic diseases with associated skeletal muscle weakness implicate genes directly related to sarcomere structure, few studies have characterized changes to sarcomere organization besides large-scale changes (i.e., sarcomere length). This is despite emerging data demonstrating that nanoscale changes in the organization of sarcomere proteins can significantly alter sarcomere function [17,26,36,48,49]. Although there are some studies that include measures of sarcomere structure in disease, there is a dearth of data relating to the structure of nanoscale sarcomere regions or protein-specific localization [17,31,50,51].

Our understanding of nanoscale sarcomere structure has been limited due to the resolution constraints of light microscopy and issues with current methodological approaches [52–56]. For example, numerous proteins localize within the Z-disc, a 30-140nm lateral space depending on the muscle and species [5,57]. Even within this nanoscale domain, there is a region-specific pattern to the localization of specific proteins [17,58–64]. Commonly used confocal microscopy cannot resolve region specific localization of Z-disc proteins, resulting in the often-vague "Z-disc localizing" characterization of proteins. Additionally, common histological approaches are challenged by the introduction of structural artifacts as well as the loss of secondary antibody specificity due to issues with traditional IgG antibodies [65–72].

In this study, we present an approach that 1) preserves sarcomere structure, 2) includes a simple, reproducible image processing pipeline for analysis of sarcomere structure, 3) can utilize multiple same-host primary antibodies species with retained secondary specificity, and 4) employs structured illumination super resolution microscopy (SIM) with emergent nanobody technology to allow for the clear localization of multiple proteins within a nanoscale space. These results are the first demonstration, to our knowledge, of the use of SIM to obtain accurate measures of Z-disc width in mouse skeletal muscle cryosections like those reported using electron microscopy. These methods, while designed for the study of sarcomere structure in preclinical rodent models, have potential for the application to other biomedical disciplines investigating the nanoscale structure and organization of biological tissues.

## Results

### Optimization of skeletal muscle preparation allows for the generation of longitudinal cryosections with preserved sarcomere structure compared to traditional approaches

To evaluate structural integrity of longitudinal muscle sections, we developed a process to standardize muscle preparations by modification of steps reported by Glancy et al., 2015 (For details, see Materials and methods) [73]. We obtained longitudinal cryosections from either fixed or non-fixed (from the contralateral leg) TA muscles and labeled them with antibodies specific to the proteins α-actinin, myomesin, and a titin-specific epitope (the titin MIR domain) for visualization of the sarcomere Z-disc, M-line, and a point of reference in between the Z-disc and M-line. In Fig 1, we provide images obtained via confocal microscopy between the fixed and non-fixed muscles demonstrating that structural regularity of all three sarcomere markers were best maintained in fixed TA sections. We analyzed the antibody labeling to

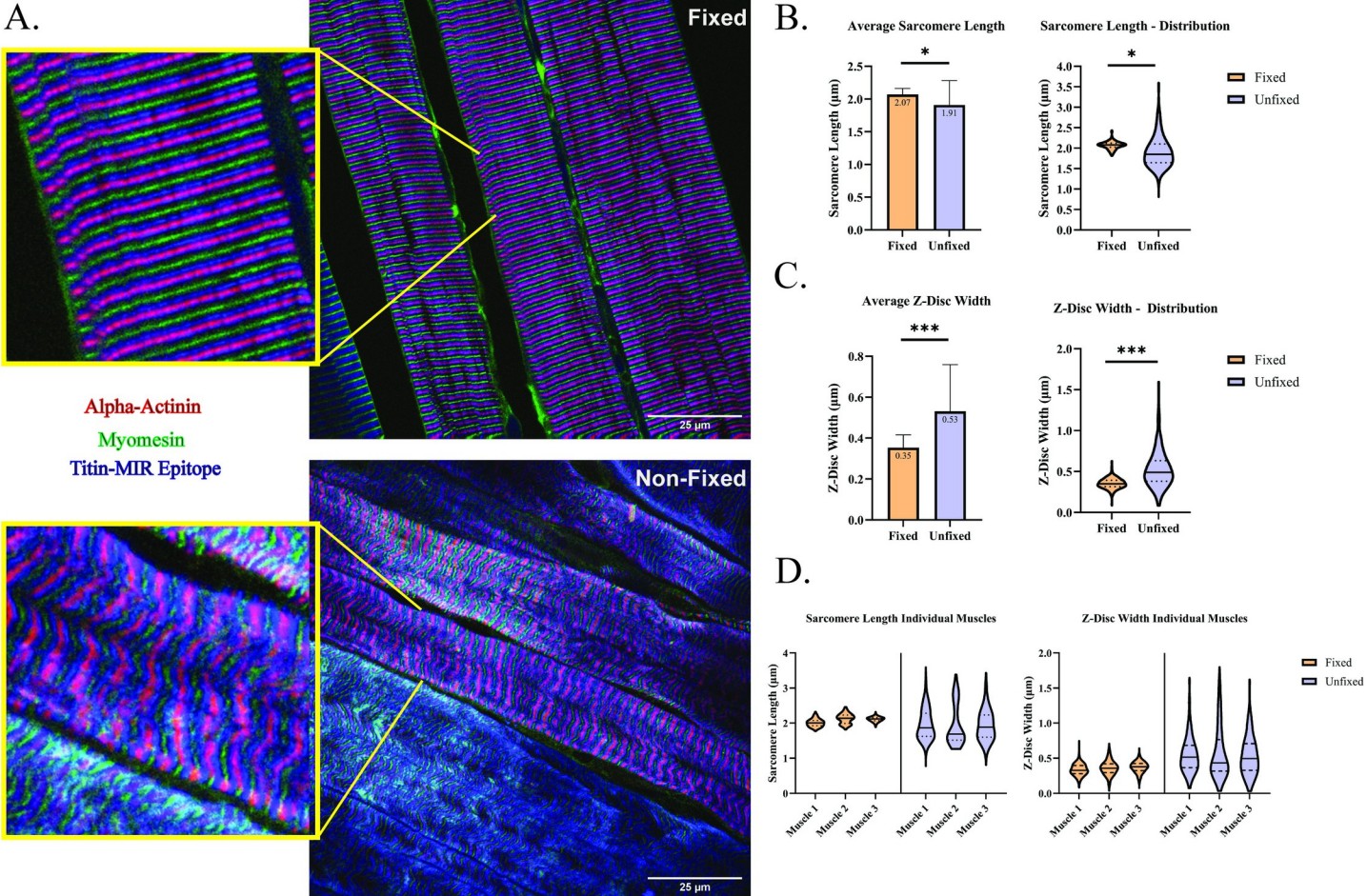

**Fig 1. Fixation process prior to embedding muscle for cryosectioning visually and quantitatively preserves sarcomere structure.** (A) Representative images of fixed (top) and non-fixed (bottom) TA cryosections labeled with antibodies specific to α-actinin (rabbit polyclonal, Z-disc; red), myomesin (mouse monoclonal, M-line; green), and the titin-MIR domain (chicken polyclonal, sarcomere A/I-band interface; blue). Enhanced images are shown highlighted with yellow border. (B) Quantification of average sarcomere length between fixed and non-fixed TA sections. (C) Quantification of Z-disc width between fixed and non-fixed TA sections (D) Violin plots demonstrating distribution of values obtained from individual biological replicates. Quantifications are from 5 non-overlapping images acquired from n = 3 biological replicates per group. Solid bars represent median values, with dashed lines indicating upper and lower quartiles. Plotted measures in all figures are shown as mean of total measures with standard deviation, statistical comparisons shown are between group means, ***$p<0.001$, * $p<0.05$. Images obtained using confocal microscopy under 60x oil-immersion objective (See *Methods*).

further assess sarcomere structure in our optimized tissue preservation process through quantitative analysis. Analysis between groups demonstrated that both average sarcomere length and Z-disc width were less homogeneous in non-fixed tissue than fixed tissue (Fig 1). Additionally, this was consistent among the measures per biological replicate demonstrating consistency between individual muscle preparations (Fig 1). While confocal microscopy does not have the resolution to obtain accurate measures of Z-disc width, the preserved homogeneity of Z-disc width, with average (Fig 1) or within a muscle (Fig 1) suggests better preserved Z-disc structure using our approach. We compared previously published *in vivo* measures of sarcomere length homogeneity (standard deviation and coefficient of variation) to compare with sarcomere length variation using our approach [7]. We found that the average standard deviation and coefficient of variation (%) were like the *in vivo* imaging, and results from the fixation

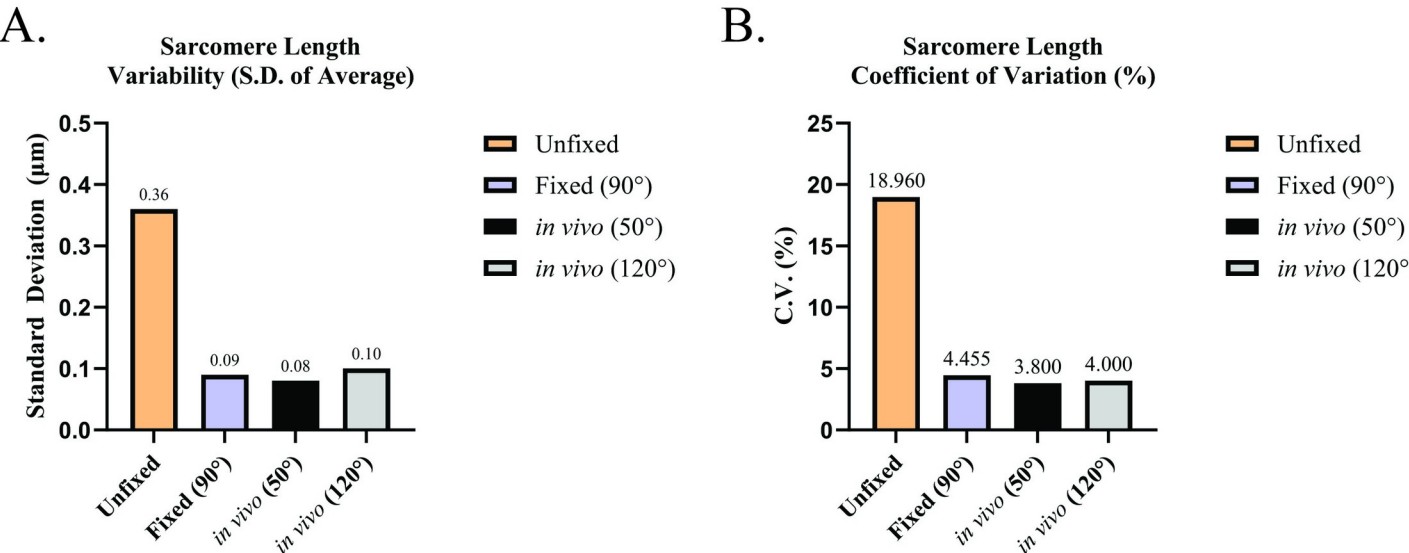

**Fig 2. Comparison of sarcomere length homogeneity measures performed in this study to literature measures from *in vivo* imaging.** Measures of sarcomere length homogeneity from this study were plotted to compare to *in vivo* data reported in Moo et al. 2016. (A) Standard deviation (SD) of average sarcomere length measures. (B) Coefficient of variation (CV) of average sarcomere length measures. Data plotted is representative of the SD or CV obtained from the average sarcomere length measures obtained in this study ("Unfixed" or "Fixed (90˚"), or of SD and CV values from "Deep Sarcomere" measures from Fig 5 in Moo et al., 2016 ("*in vivo* (50˚)" or "*in vivo* (120˚)"). Angles (˚) in figure legend refer to the relative angle between foot and tibia used during measurements.

of the muscle tissue (Fig 2). These comparative data demonstrate the efficacy of our approach for maintaining sarcomere structure during the muscle preparation process.

## Use of unconjugated FAB fragments allows for multiple same-host primary antibody labeling without loss of secondary antibody specificity

A common issue that arises using immunofluorescent analysis is that many commercially available primary antibodies are commonly sourced from the same host species. This makes localization analysis of multiple proteins within an image challenging due to potential loss of secondary antibody specificity. Recently, IgG antibody fragments—Fragment Antigen Binding antibodies, or FAB antibodies—with no additional conjugation have been developed and suggested for use to block residual antigen binding sites as a method to allow for the use of multiple same-host primary antibodies with retention of secondary antibody specificity [74]. We tested this application by utilizing three different rabbit-derived primary antibodies (α-actinin, the titin C-terminus, and the titin-MIR domain) that are specific to three spatially distinct sarcomere regions that localize within ~1μm—the Z-disc, the M-line, and the sarcomere A/I-band interface, respectively (Fig 3). Without use of FAB antibodies or temporal segregation of antibody labeling, we observed an expected overlap of two individual secondary antibodies targeting two distinct primary antibodies (Fig 3). By temporally segregating antibody labeling, but without FAB antibodies, protein labeling was qualitatively more specific; however, there were still considerable instances of overlap between secondary antibodies (Fig 3). Using the same labeling sequence, but with addition of FAB antibodies, we observed little-to-no overlap of secondary antibodies, allowing for definitive localization of two primary antibodies derived from the same host-species (Fig 3). We further tested this approach through the labeling of three distinct primary antibodies with the addition of FAB antibodies, and clear localization of each primary antibody was maintained (Fig 3). We additionally noted no off-target labeling of

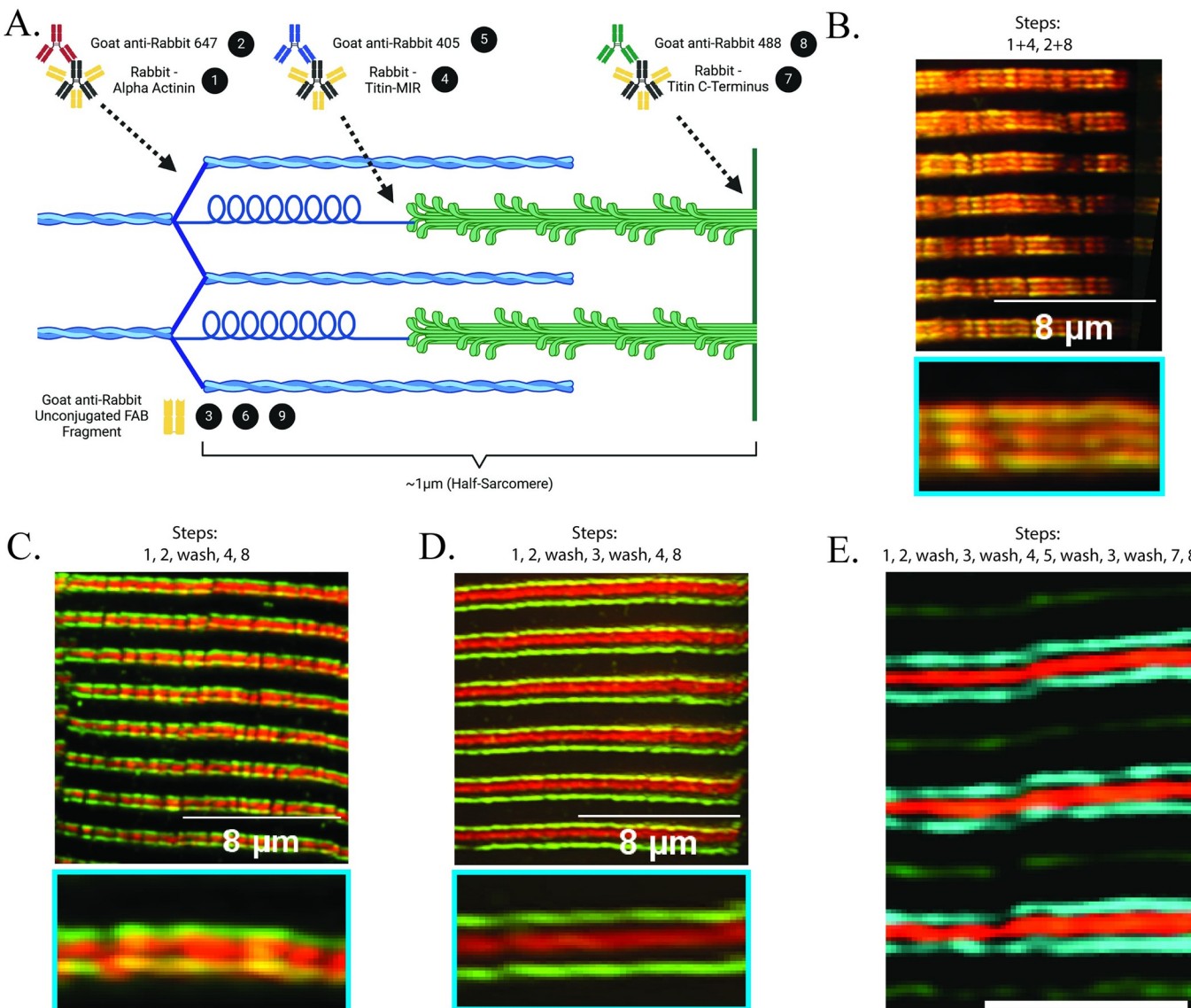

**Fig 3. Utilizing a FAB antibody post-blocking step prior to addition of subsequent same-host primary antibodies significantly limits potential overlap of secondary antibody labelings.** (A) Cartoon (produced with BioRender; not to scale) demonstrating an example of the labeling scheme used which directly corresponds to labeling sequence used in panel E. Black circles refer to the specific steps in the protocol outlined in *Materials and Methods*. (B) Representative image showing result of applying two same-host primary antibodies at the same time followed by simultaneous incubation with 488- and 647-conjugated secondary antibodies. (C) Representative image demonstrating application of two same-host primary antibodies, followed by incubation with 488- and 647-conjugated secondary antibodies, but with applications temporally segregated. (D) Representative image of two same-host primary antibody incubations including post-blocking steps using anti-rabbit unconjugated FAB antibodies prior to subsequent antibody incubations. (E) Representative image of TA section labeled triple-labeled with same-host primary antibodies corresponding to steps in panel A. Text above individual panels corresponds to specific antibody incubations (black circles with numbers in panel A) and/or wash steps ("wash") used in the labeling protocol; sequential steps are separated by a comma, whereas simultaneous steps are grouped with a "+" (See Materials and methods for detailed protocol). Images obtained using SIM under 60x oil-immersion objective (See Materials and methods).

subsequent secondary antibodies following blocking with FAB antibodies through inspection of individual image channels (S1 Fig). These observations demonstrate the efficacy of FAB antibodies as a resource to maintain secondary antibody specificity for experiments using up to three same host-species antibodies.

## Development of an image analysis pipeline to decrease subjectivity in measures of sarcomere structure

Line scan analysis is a commonly used approach to measure sarcomere length, or the relative distance between sarcomere proteins (Fig 4). However, this approach is subject to user-introduced bias and variability based on the point of reference used for measurements (Fig 4). Fast Fourier Transforms (FFT) have been used to automate these measures but the accuracy of FFT requires healthy, regular sarcomeres and may inappropriately estimate these measures when there are structural disruptions, such as in disease or weakness [75]. Therefore, we developed an image analysis pipeline to refine immunofluorescent images of specific sarcomere protein labeling for use with line scan analysis to reduce user bias and variability (Fig 4; see Materials and methods for details) [76]. In brief, this pipeline shown in panel A of Fig 4 consists of:

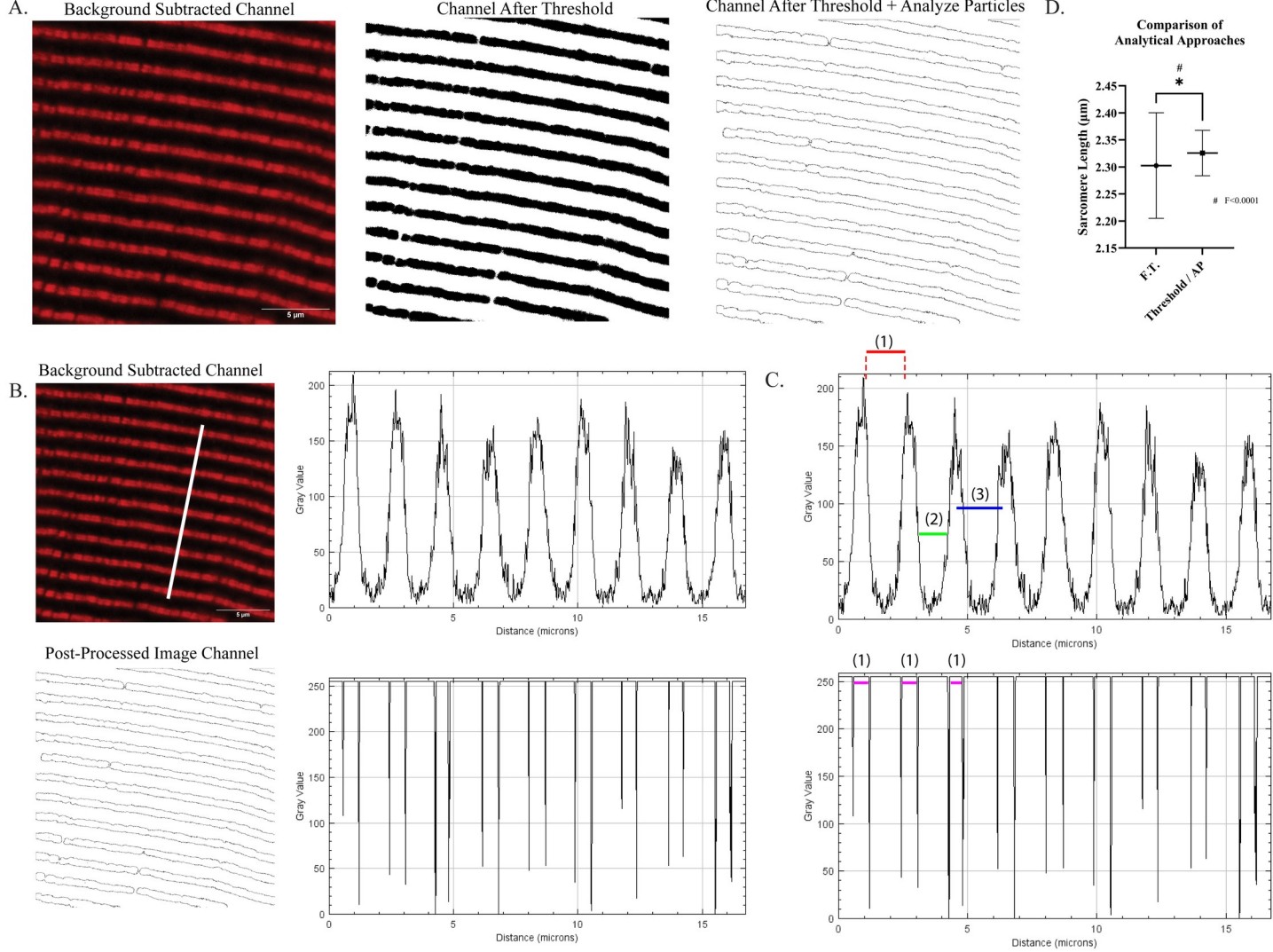

**Fig 4. Demonstration of a simple, reproducible approach to generate specified points of reference corresponding to fluorescently labeled proteins through image pre-processing in Fiji software.** (A) Example images of a TA cryosection labeled for an antibody specific to the N2A domain of titin with average background subtracted (left, in red), binarization applied (middle), and particles analyzed (right). (B) Example images with corresponding plot profiles demonstrating difference in pre- and post-processed images. (C) Demonstration of differences in points of reference for measures of sarcomere structure between pre- and post-processed images. (D) Comparison of sarcomere length analysis of identical line scans using either Fast Fourier Transform (F.T.) or optimized approach (Threshold/AP). * Student's t-test p<0.05, # F-test p<0.0001 Images obtained using confocal microscopy using 63x oil-immersion objective (See Materials and methods for details).

- Background subtraction per individual image channel

- Binarization of the image, promoting foreground (fluorescent label) and further eliminating background per individual channel

- Particle analysis to clearly delineate borders of the fluorescent signal per individual channel

A comparison of approaches is provided in panel B of Fig 4; the unrefined image with its corresponding line scan results in a corresponding profile plot with undefined, broad peaks while our pipeline results in defined lines indicating the border of the fluorescent label. Introducing the particle analysis removes the requirement for subjectively assigning points of reference and results in consistently and automatically assigned points of reference, as demonstrated by the colored lines in panel C of Fig 4. We tested the efficacy of this pipeline to obtain accurate measures of sarcomere length compared to automated FFT analysis. Comparing identical line scans across the same images, we found that FFT analysis significantly under-represented average sarcomere length and was significantly more variable compared to our optimized line scan analysis approach. Importantly, the homogeneity of the measures obtained after utilizing our image pre-processing pipeline were more like measures of sarcomere length homogeneity obtained via *in vivo* imaging approaches compared to FFT analysis (Figs 2 and 4). By utilizing this pipeline across multiple channels of the same image, measures of multiple sarcomere protein localizations can be performed in a consistent manner through the assignment of clear, objective points of reference per fluorescent label.

## Utilizing super-resolution microscopy in concert with emergent nanobody technology significantly enhances the ability to accurately localize sarcomere proteins at the nanoscale

To gain higher resolution for analysis of nanoscale sarcomere structure and protein localization, we moved our approach from confocal microscopy to structured illumination super resolution microscopy (SIM) and tested the ability of SIM to 1) obtain accurate measures of Z-disc width using immunofluorescence and 2) delineate between multiple protein-specific antibodies within nanoscopic proximity. SIM provides multiple-fold improvements in resolution above confocal microscopy [5,52,77]. We acquired images of TA sections fluorescently labeled with antibodies specific to the Titin N2A domain to test for improvements in resolution with SIM. TA muscle sections were labeled for the N2A domain of Titin and imaged with either confocal microscopy or SIM (Fig 5). As seen in Fig 5, SIM accurately resolved two regions of N2A-Titin labeling on either side of the Z-disc, while these two distinct regions could not be resolved with confocal microscopy. We next tested the ability of SIM to accurately measure the width of the Z-disc. Analysis of sections labeled with antibodies specific for the Z-disc protein, α-actinin, and imaged with either confocal microscopy or SIM demonstrated that Z-disc width was qualitatively and quantitatively narrower when imaged with SIM (Fig 6). Importantly, these measures were much closer to reported measures of Z-disc width obtained with electron microscopy [5,57].

Another opportunity for optimization that we identified is with the use of newly developed and commercially-available fluorescent secondary nanobodies. The accuracy of protein localization of immunofluorescent microscopy has historically been limited by antibody linkage error—the molecular distance between the protein of interest and the fluorophore used for visualization [78–82]. We next tested the efficacy of fluorescently-conjugated Variable Heavy-Chain (VHH) only fragment secondary antibodies, or nanobodies, which are substantially smaller in size than traditional IgG antibodies, to enhance the protein localization accuracy of

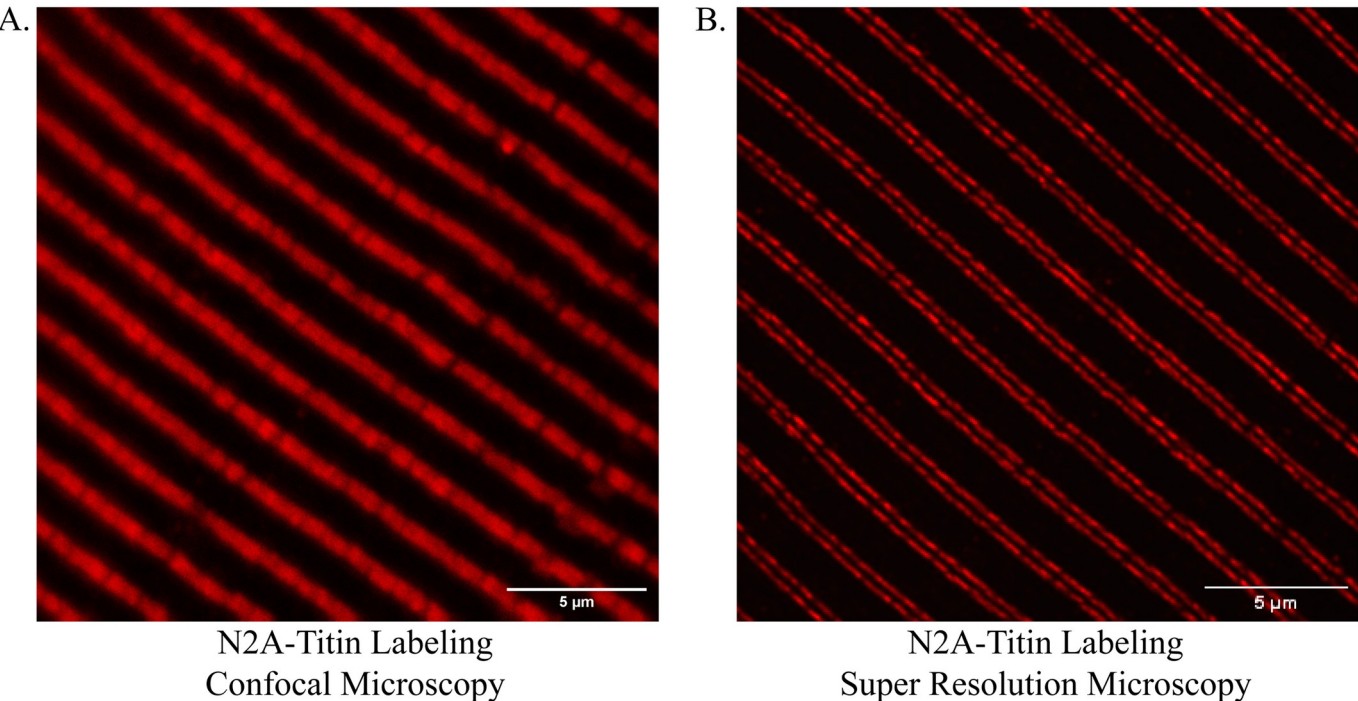

N2A-Titin Labeling
Confocal Microscopy

N2A-Titin Labeling
Super Resolution Microscopy

**Fig 5. Representative images demonstrating increased nanoscale resolving capability using SIM compared to confocal microscopy.** (A) Representative image of TA cryosection labeled with antibody specific to the Titin N2A domain and imaged with confocal microscopy. (B) Representative image of TA cryosection labeled with antibody specific to the Titin N2A domain and imaged with SIM. Images obtained using either confocal microscopy under 63x oil-immersion objective or SIM under 60x oil-immersion objective (See Materials and methods).

SIM and obtain accurate measures of Z-disc width [78,81,82]. Using traditional IgG antibodies and comparing Z-disc width measures obtained used either confocal or SIM, there was an expected and observed decrease in Z-disc width measures with SIM because of the increase in optical resolution (Fig 6). However, we additionally observed a significant decrease in Z-disc width measures using SIM in concert with nanobodies compared to traditional IgG (Fig 6). These data demonstrate that the use of nanobodies with SIM results in a significant improvement of protein localization accuracy as noted by a decrease in Z-disc width measures compared to IgG antibodies.

We collectively assessed the robustness of our optimized approach to localize multiple proteins within a nanoscale proximity within the sarcomere. Analysis of sections labeled with nanobodies targeting three protein epitopes (α-actinin, Titin-N2A, and Titin-PEVK) with known localizations within a ~200nm lateral space and imaged using SIM demonstrated a clear delineation of respective proteins (Fig 6) [13,19,83]. This proof-of-concept example effectively demonstrates that these collective protocols can resolve multiple sarcomere protein localizations within a nanoscale domain.

## Discussion

Skeletal muscle weakness is clinically relevant, as it is a common comorbidity across various chronic diseases [20,34,39–45,84]. Further, epidemiological studies have demonstrated that decreased skeletal muscle strength is a better predictor of increased mortality than loss of skeletal muscle mass [34,37,39,84]. It is also known from both preclinical and clinical studies that muscle weakness commonly precedes the loss of skeletal muscle mass [84–87]. Collectively,

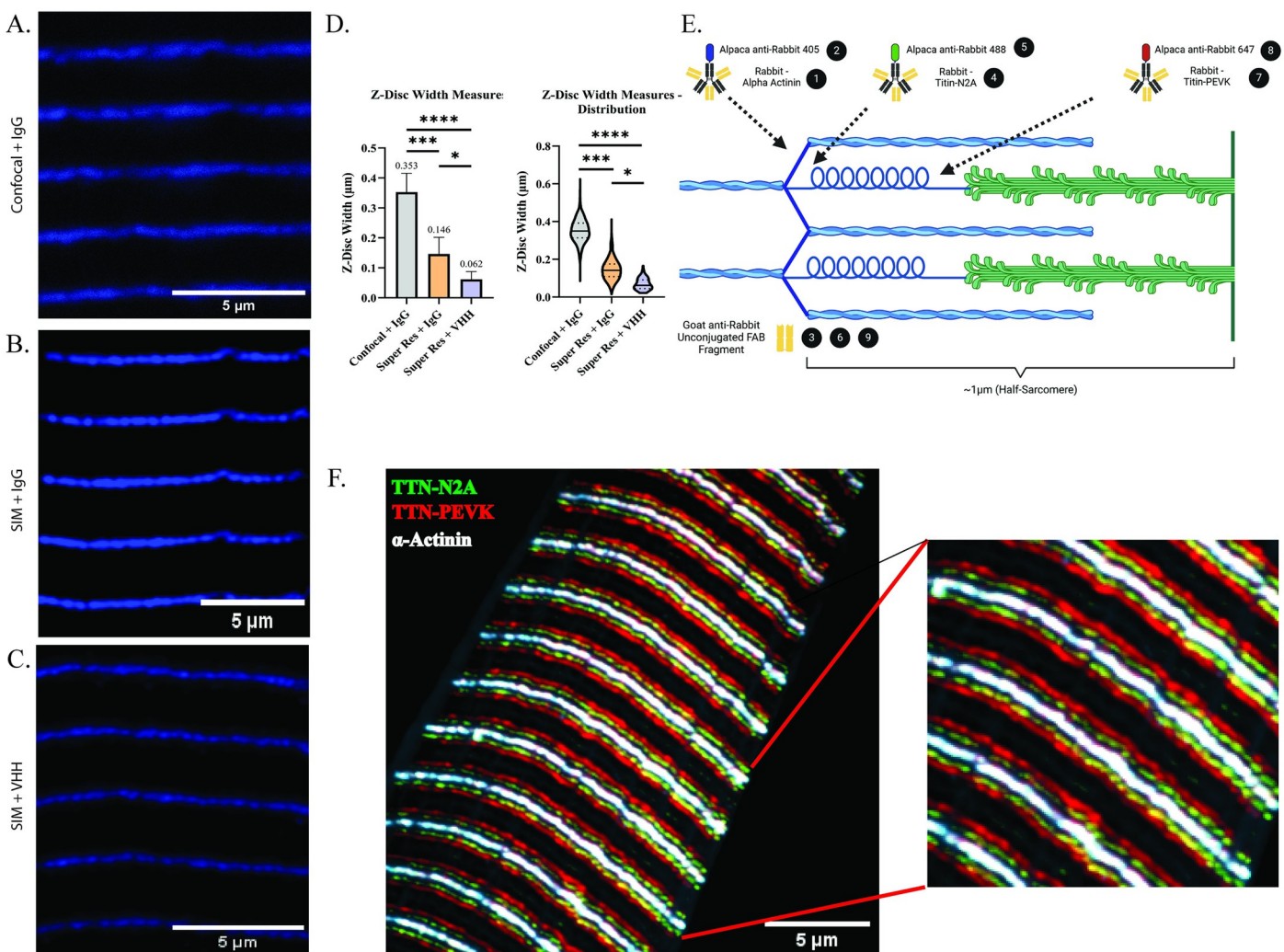

**Fig 6. Utilization of fluorescently conjugated nanobodies enhances nanoscale protein localization accuracy of SIM.** (A,B,C) Representative images of TA cryosections labeled with an antibody specific to α-Actinin prior to incubation with (A,B) traditional IgG or (C) VHH secondary nanobodies and imaged using either (A) confocal microscopy or (B,C) SIM. (D) Quantification of Z-disc width measures obtained from confocal microscopy, SIM, and SIM + VHH secondary nanobodies. (E) Cartoon (not to scale, made with BioRender) depicting labeling scheme used for images in panel F. Numbers in black circles represent individual antibody incubations as in *Materials and Methods*. (F) Representative image demonstrating clear localization of three targeted primary antibodies (α-Actinin, Titin-N2A, Titin-PEVK) with nanobody secondaries within a nanoscale proximity. Images obtained using either confocal microscopy under 63x oil-immersion objective or SIM under 60x oil-immersion objective. Quantifications were performed using a minimum of n = 3 biological replicates per group, 5 non-overlapping images per biological replicate, and 5 non-overlapping line scans per image. Violin plots demonstrate the distribution of measures obtained per experimental group. Statistical comparisons were made using one-way ANOVA analysis, **** p<0.0001, *** p<0.001, * p<0.05.

these studies implicate intrinsic changes to the skeletal muscle prior to loss of mass that are responsible for the observed decline in strength. One subcellular feature of muscle that can contribute to declines in strength is the structural integrity of sarcomeres—the individual, force generating unit in skeletal muscle [8]. Despite the importance of sarcomere structure in muscle function, our understanding of how sarcomere structure is specifically affected in models of muscle weakness has been limited by lack of robust and reliable immunofluorescence methods. Additionally, the methods used for the analysis of sarcomere structure in both clinical and pre-clinical research have not incorporated emergent imaging modalities and analytical approaches. Thus, there is a need for the modernization of sarcomere structure analysis in

pre-clinical models which includes optimization of proven approaches in concert with the incorporation of emergent technologies.

Important sarcomere regions such as the Z-disc and M-line, as well as specific domains of the giant protein Titin have been linked to the regulation of muscle force [10,17,19,48,57,88–90]. It has been well established that nanoscale changes in the organization of the sarcomere can impact sarcomere force output [26,91]. Despite this recognition, our understanding of the nanoscale organization of these domains and proteins is arguably still in its infancy [26,59,91–93]. Historically, the approaches used have included *ex vivo* light microscopy, electron microscopy, immunohistochemistry, and immunofluorescence with tissue sections or isolated myofibers, and more recently *in vivo* imaging modalities [7,25,65–72,94–101]. These approaches have unique strengths and limitations, but the most notable delineating factors between the *ex vivo* and *in vivo* approaches are largely differences in 1) the preservation of sarcomere structure, 2) the ability to label multiple proteins, and 3) the resolution provided. For example, *in vivo* imaging provides images of true sarcomere structure, while tissue processing necessary for *ex vivo* approaches often disrupts sarcomere structure that is evident in the resulting images [7,65–71,94,96,101]. Conversely, *in vivo* imaging is limited in its inability to label specific proteins, whereas *ex vivo* approaches retain this ability; among the *ex vivo* approaches, immunofluorescent labeling excels over immunogold labeling with respect to localization analysis of multiple labeled proteins within the same histological section. Finally, electron microscopy has historically been lauded as the gold-standard approach for nanoscale cellular analysis. However, the advent of immunofluorescent super resolution imaging modalities has expanded the feasibility of nanoscale cellular image analysis. Therefore, in this study we developed a robust and reproducible methods pipeline to analyze nanoscale sarcomere structure via SIM in concert with commercially available nanobody technology, while using longitudinal cryosections with preserved sarcomere structure. While this approach was designed for use in skeletal muscle research, some aspects of these methods may have application to other cell and tissue imaging research.

In this study, we developed a methodological pipeline using a standardized muscle preparation that allows for reproducible, quantitative measures of sarcomere structure that are more representative of *in vivo* data [7]. Commonly used protocols across the literature do not maintain muscle length during the tissue preparation process. The results presented determined significant improvement in sarcomere length and variability measures that align with outcomes from *in vivo* analysis [7]. We want to note that the attention to length when preparing muscle for histological analysis impacts more than just longitudinal structure analysis. Skeletal muscle is an isovolumic tissue, meaning that changes in width impact length and vice-versa. We argue that lack of attention to length also impacts the ability to compare measures of muscle fiber cross-sectional area across studies and labs. This issue is highlighted by a recent study that compared muscle and muscle fiber cross-sectional area measures across studies and found that there was limited consensus of human muscle fiber cross-sectional area, likely confounded by limited attention to muscle length during muscle preparation [102]. Additionally, we note that there are available approaches designed to automate measures of sarcomere structure [103,104]. While these approaches are comparatively advantageous due to their automated approach, these approaches have often been optimized to investigate the myofibrillar organization of cardiomyocytes in culture. A common issue involved in the study of myofibrillar organization of cultured myocytes is that myofibrillar organization is highly influenced by *in vitro* cell culture conditions and is therefore not representative of sarcomere structure visualized through *in vivo* or *ex vivo* imaging approaches [105–109]. Further, many automated methods utilize fast Fourier transform approaches to obtain these measures. Recent data suggests that this approach is limited in its ability to accurately perform these measures, especially in disease

models in which myofibrillar organization may be altered [75]. Collectively, these issues undercut the ability to appropriately compare data across experimental models and models of skeletal muscle disease. We note that the approach presented herein would benefit from automation, but it provides comparative evidence to demonstrate that it produces measures that are less variable than traditionally used fast Fourier transform analysis but also more like *in vivo* measures than other common approaches. We hope that this approach may provide a foundation to further optimize sarcomere structure analysis regardless of experimental conditions.

The next development was a simple protocol that allows for the labeling of up to three proteins on the same histological section using IgG antibodies derived from the same host-species. This is important considering the dichotomy between the need to label multiple proteins to determine relative changes in sarcomere organization and the associated issue of limited species used for commercial antibody production. Within this protocol we additionally demonstrate the efficacy of utilizing emergent nanobody technology to decrease antibody linkage error with quantitative evidence of enhanced protein localization accuracy in concert with SIM via measures of Z-disc width that align with electron microscopy data. The use of nanobodies additionally allows for the accurate delineation of multiple targeted antibodies within a nanoscale space, allowing for the analysis of multiple sarcomere proteins within the same histological section. We demonstrate this ability with a proof-of-concept example by labeling and imaging the Z-disc and two specific epitopes of the protein Titin (the N2A and PEVK domain). By doing so, the distance between N2A domains across the Z-disc (~570nm) as well as between the N2A and PEVK domains (~170nm) can be estimated (S2 Fig). While data estimating these distances is sparse, the estimates provided in this study are smaller than prior measures obtained from isolated human diaphragm myofibers using the same imaging modality at a similar sarcomere length (N2A-N2A ~570nm vs. ~600nm; N2A-PEVK ~170nm vs. ~210nm estimated) [19]. We would like to note, and appreciate, that previous studies have used super-resolution single-molecule localization microscopy (STORM and DNA-PAINT) to analyze Z-disc structure in *Drosophila* flight muscles [110,111]. However, these studies utilized either single myofibril isolation methods which can be technically cumbersome compared to the acquisition of cryosections, or do not report measures of Z-disc width. Further, single-molecule localization microscopy techniques such as STORM and DNA-PAINT require a greater amount of technical expertise compared to SIM due to the requirement of specialized fluorophores, imaging reagents, and conditions, along with the time spent for optimization of these components [112]. In addition, these imaging modalities have specific technical challenges associated with multi-color labeling, creating an additional issue with respect to the coincident analysis of multiple sarcomere proteins within the same sample. Thus, we believe that the combination of approaches demonstrated in the current study provides a means to analyze sarcomere structure in pre-clinical mouse models that is more economical, with respect to both time and material cost, than prior methods. Lastly, we designed an image pre-analysis refinement approach to reduce the potential for subjectively assigned points of reference for protein localization and demonstrate its efficacy to reduce variability in measures of sarcomere structure. *In vivo* data demonstrates that homogeneity of sarcomere length is a robust characteristic of healthy skeletal muscle tissue. The results with our methods demonstrated measures of sarcomere length variability that align with *in vivo* measures, and electron microscopy data [7].

For skeletal muscle research, these methods provided will allow for greater in-depth analyses of protein localization within sarcomeres. For example, many sarcomere proteins reside within distinct sub-regions, such as the Z-disc or M-line, but there are multiple proteins within those sub-regions. Well recognized examples include Tcap and Muscle LIM Protein within the Z-disc and Murf1 within the M-line [113–115]. The methods described will provide new

capabilities to assess changes in the localization of these proteins in models of disease and weakness. There is growing recognition of the importance for the specific roles of these proteins within these nanoscale domains such as modulation of muscle function, signaling, or sarcomere region-specific structure [5,17,19,23,48,49,57,59–62,64,80,89,116–119]. Advances in the ability to accurately localize proteins will allow a greater number of researchers to characterize the nanoscopic specificity of protein localization within these important domains (i.e. central, or peripheral Z-disc localization), or with respect to other proteins (i.e. localization of proteins relative to a specific domain of Titin, or specifically where Titin's N2A domain interacts with the Actin filament, etc.) and thereby offer a deeper understanding of sarcomere organization in health and disease.

## Conclusions

The well-defined organization of proteins within sarcomeres is critical for the maximal force-producing capabilities and thus, strength, of the muscle. In addition, proteins important for signaling in response to increased muscle use or dis-use reside within the sarcomere. Further, even nanoscale changes in the localization of these proteins may impact their function or the function of the sarcomere. However, our understanding of the nanoscale localization of these proteins, or how those localizations change in disease or weakness, is limited by commonly used approaches that are not modernized with emerging technologies. In this study, we developed and presented a robust and reproducible approach to analyze nanoscale sarcomere structure using longitudinal sections of mouse TA muscle. This approach qualitatively and quantitatively maintains sarcomere structure representative of data from *in vivo* imaging, allows for the accurate localization of numerous targeted proteins within a nanoscale domain using emergent nanobody technology, and implements an image pre-analysis refinement pipeline to obtain measures of sarcomere structure using immunofluorescence that are more like both *in vivo* and electron microscopy data than commonly used approaches across the literature. The methods presented in this study provide a robust toolkit for the study of nanoscale sarcomere structure, as well as provide tools and resources for the nanoscale analysis of cellular biology in other disciplines of biomedical research.

## Material and methods

### Ethical approval

All animal work performed in this study was conducted in accordance with and approved by the Institutional Animal Care and Use Committee at the University of Florida (IACUC #202009900). The use of animals for all experiments was in accordance with the guidelines established by the US Public Health Service Policy on Humane Care and Use of Laboratory Animals.

### Animals and tissue collection

5 male C57BL6 mice (7–10 months of age) were used for all experiments. Mice were housed under a 12:12 light-cycle corresponding to lights on at 7am and lights off at 7pm, with *ad libitum* access to standard rodent chow (Envigo Teklad 2918, Indianapolis, IN, USA) and water. Mice were anaesthetized using isoflurane followed by secondary euthanasia via cervical dislocation as approved by UF IACUC. Tissue collections were performed at a similar time daily (~3pm) to avoid any potential time of day effects in muscle.

## Muscle processing for histological sectioning

Process of muscle preparation for fixation and downstream cryosectioning is seen in Fig 7 (modified from work by Glancy et al., 2015) [73]. Entire legs of mice had skin completely removed before being fully removed above the knee. After removal, legs were placed on the flat side of a cork that was cut in half through the middle, lengthwise, with the tibialis anterior (TA) muscle facing upwards away from the cork to better expose it to fixative. TA muscle was chosen since it is a commonly used muscle in the study of sarcomere structure in pre-clinical rodent models. The legs were pinned in place with needles through the remaining tissue above the knee, one through the gastrocnemius muscle, and one through the mid-foot with the knee and ankle joints fixed in place at 90° relative to the tibia to standardize muscle length. This step was critical since changes in muscle length by way of changes in the angle of the knee and ankle joints can affect sarcomere length [120]. Corks with leg attached were then placed in a 50mL screw-top conical tube and completely submerged in 10% neutral-buffered formalin (Fisher Healthcare, USA; Cat #23–305510) and left overnight at 4°C. The following day,

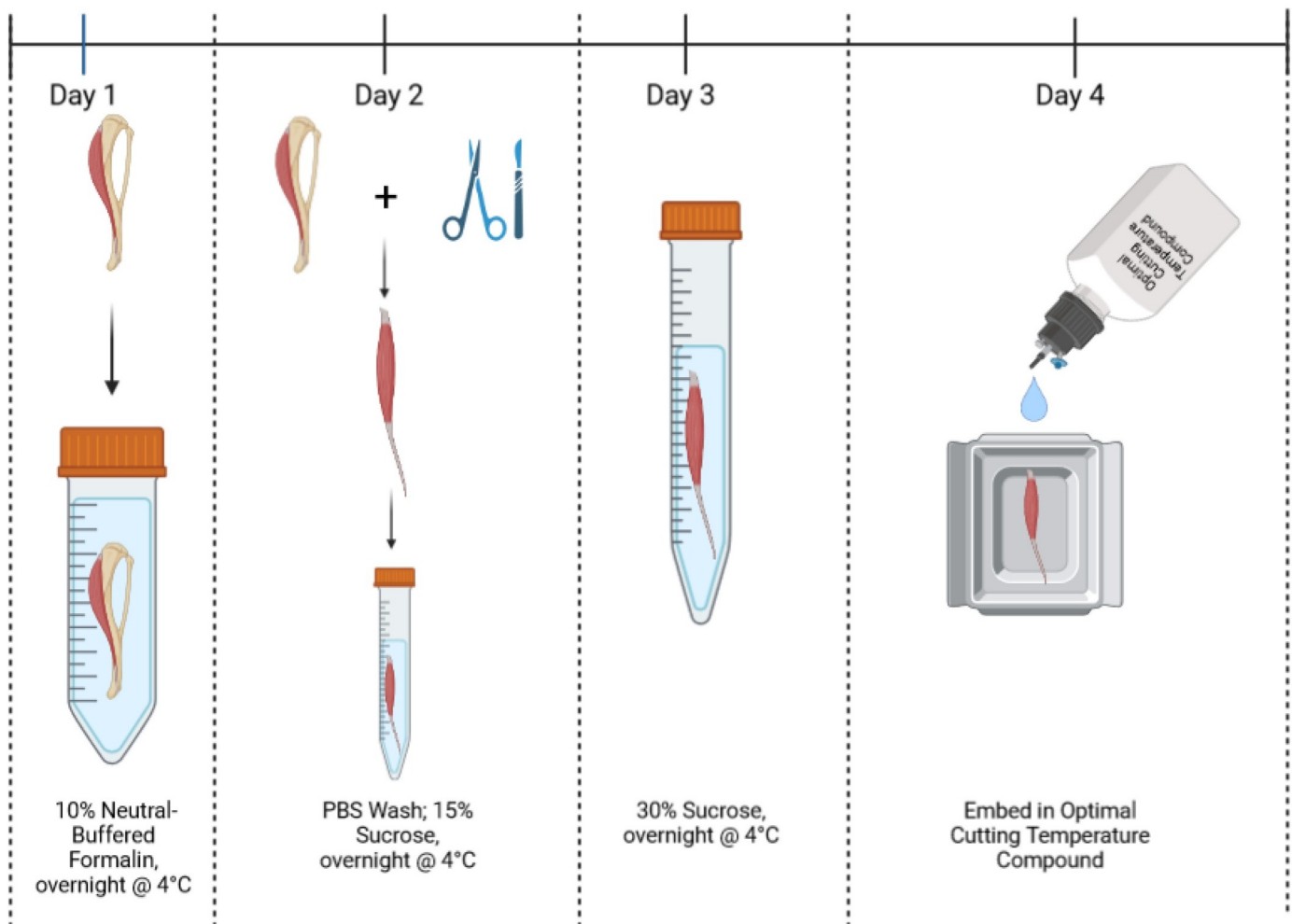

**Fig 7. Cartoon demonstrating process of muscle fixation prior to preparation for longitudinal sectioning.** In brief, the mouse leg with the tibialis anterior muscle still attached is skinned and removed. The leg is placed on a piece of cork with knee and ankle joints fixed at standardized angles. The leg is then placed in 10% neutral-buffered formalin overnight. The following days, the muscle is removed, washed, and cryoprotected through a sucrose gradient. It is then frozen in OCT compound for cryosectioning. Cartoon generated using BioRender.See Materials and methods for details.

formalin was replaced with Tris-buffered saline (TBS) solution to wash for 5 minutes. Following this brief wash, the TA muscle is carefully removed from the leg, placed in TBS for an additional 5 minutes before transfer to a 15mL screw-top conical tube filled with 15% sucrose solution (in TBS), and placed overnight at 4˚C. The following day, 15% sucrose solution was removed and replaced with 30% sucrose solution, and placed overnight at 4˚C. The following day, the TA muscle was placed in TBS for 5 minutes before being placed on a kimwipe to absorb residual moisture. The TA was then placed in Tissue-Tek O.C.T. Compound (O.C.T.; Sakura Finetek, USA; Cat #4583) and frozen in liquid nitrogen-cooled isopentane. Tissue blocks were stored in 5mL screw-top conical tubes at -80˚C until needed for sectioning and immunolabeling. TA muscles were sectioned through the longitudinal plane at an angle perpendicular to the direction of the myofibers at a thickness of 7μm. Sections for immunofluorescent labeling were obtained medially with respect to the axial plane of the muscle. TA muscles from the contralateral leg were used for control, non-fixed tissues. Control muscle length was standardized to a similar length as fixed tissue by pinning of distal and proximal TA tendons during tissue OCT embedding process.

## Immunolabeling, imaging, and antibodies used

An example of the use of multiple same-host primary antibodies labeling scheme is shown in Fig 8, with numbers in black circles showing subsequent steps relating to antibody incubations. Slides were allowed to equilibrate to room temperature (15 minutes) before sections were individually encircled using a hydrophobic PAP pen (Electron Microscopy Sciences, USA; Cat #71312). Sections were then rehydrated for five minutes using TBS. Sections were then incubated for 10 minutes with 0.5% Triton X-100 in TBS. Sections were subsequently washed for five minutes in TBS, and this wash was repeated three times. Following these washes sections were incubated for 30 minutes with one drop of Image-iT™ FX signal enhancer (Invitrogen, USA; Cat #I36933). Sections were washed with TBS for five minutes before a one-hour incubation at room temperature with blocking solution consisting of 5% normal goat serum, 5% normal alpaca serum, 5% bovine serum albumin, 1% glycine, and 0.1% Triton X-100 in TBS. Following incubation with blocking solution, sections were incubated with primary antibody diluted in blocking solution overnight at 4˚C. The next day, sections were washed for five minutes in TBS + 0.1% Tween-20 (TBS-T) three times, with a following five-minute wash in TBS before incubation with secondary antibody diluted in blocking solution for one hour at room temperature. Following secondary antibody incubation, sections were washed three times for five minutes in TBS-T, with a subsequent five-minute TBS wash. Sections were then briefly incubated (three minutes) with 4% paraformaldehyde (PFA) solution, and subsequently washed twice in TBS for five minutes each. Prior to additional primary antibody incubation, sections were next incubated for one hour at room temperature with excess unconjugated FAB antibodies (Jackson ImmunoResearch, USA) diluted in blocking buffer to block residual binding sites of primary antibodies. Following incubation sections were washed for five minutes in TBS-T three times, and subsequently washed once in TBS for five minutes. Addition of additional primary and secondary antibodies followed the same steps following the original incubation in blocking solution, for up to two additional primary and secondary antibodies. Following final incubation sections were washed for five minutes once in ultrapure water before removal of residual liquid, and mounted with ProLong™ Glass antifade mountant (Invitrogen, USA; Cat #P36980) using #1.5H coverslips (ThorLabs; Cat #CG15KH1). Antibodies used in this study along with concentrations and sources used may be found in the table below (Table 1).

Sections were imaged using a Leica SPE Confocal microscope with a separate Leica DFC7000 T Fluorescence high-speed camera under a 63x oil-immersion objective, or with a

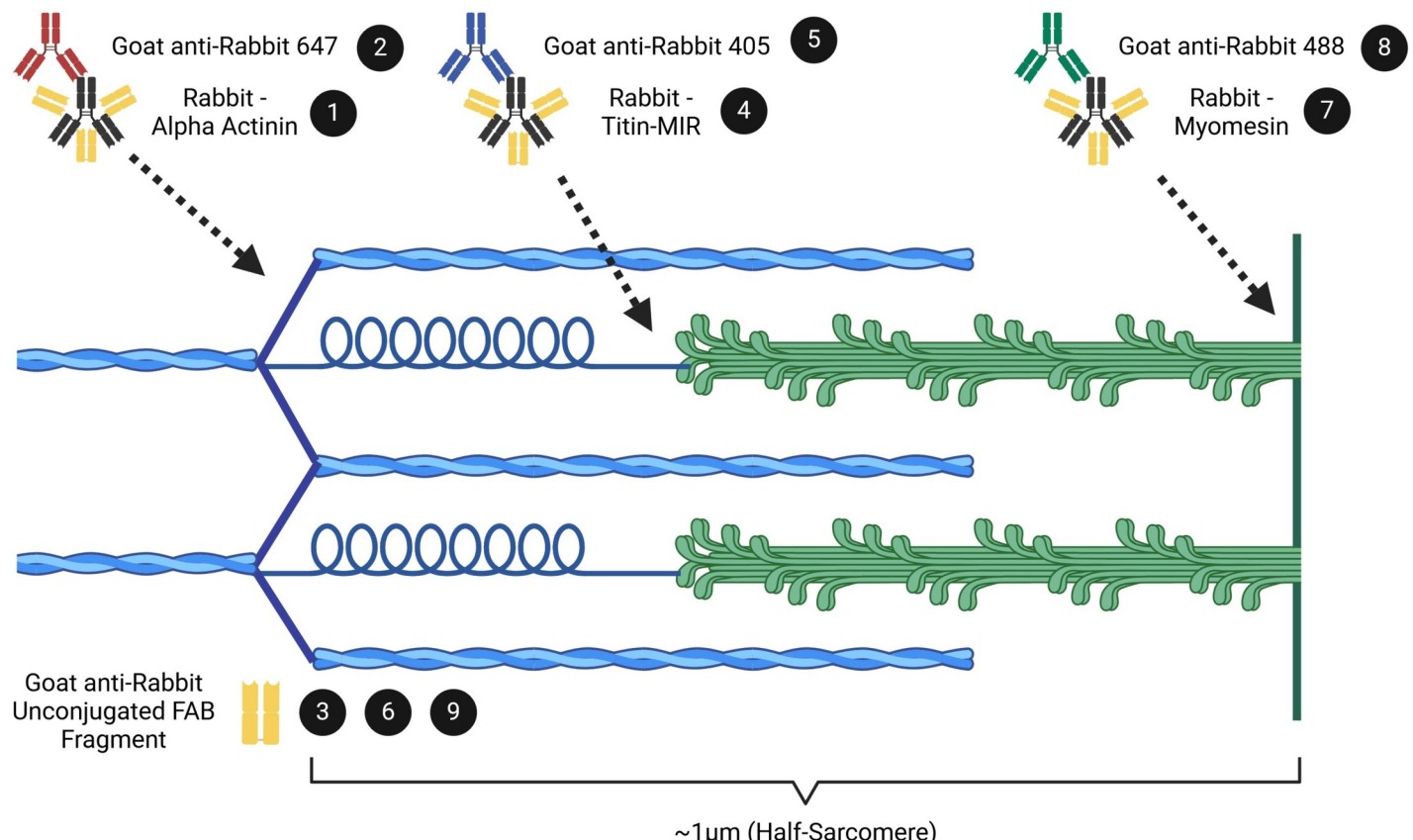

**Fig 8. Cartoon demonstrating an example labeling scheme with three same-host primary antibodies with corresponding sarcomere locations across a half-sarcomere.** Numbers in black circles correspond to the individual incubation steps listed. (1) α-Actinin antibody is used to label the sarcomere Z-disc. (2) AlexaFluor 647 secondary antibody is used to label α-Actinin antibody. (3) Anti-rabbit unconjugated FAB antibodies are used to block residual open binding sites on α-Actinin primary antibody. (4) Titin-MIR antibody is used to label the sarcomere A/I-band interface. (5) AlexaFluor 405 secondary antibody is used to label the Titin-MIR primary antibody. (6) Anti-rabbit unconjugated FAB antibodies are used to block residual open binding sites on Titin-MIR primary antibody. (7) Titin C-terminus antibody is used to label the sarcomere M-line, or middle of the sarcomere. (8) AlexaFluor 488 secondary antibody is used to label Titin C-terminus antibody. (9) Anti-rabbit unconjugated FAB antibodies are used to block residual open binding sites on Titin C-terminus primary antibody. Cartoon (not to scale) produced with BioRender.

Nikon Ti2-E inverted microscope equipped with a 60x oil-immersion objective (CFI Apochromat TIRF, NA 1.49) and a confocal spinning disk (Yokogawa, CSU-X1) with super-resolution imaging module (Gataca LiveSR) Systems) and sCMOS camera (Teledyne Photometrics, Prime95B). Z-stacks were obtained in 0.3μm increments set to 300ms exposure per individual channel and reconstructed using maximum intensity slice projections. Image analysis and processing were all performed using ImageJ software. Image brightness and contrast were adjusted using the *Auto* feature in ImageJ. Average background subtraction was performed by use of the rectangle tool to first obtain the mean gray value from a background region of interest outside of the fluorescent label per channel and was subtracted globally from individual channels. Thresholding of individual channels was performed to obtain binary images by use of the *Threshold* function in ImageJ, using the default IsoData algorithm within ImageJ, and applied automatically. Finally, the *Analyze Particles* function was used on individual binary channels to obtain Bare Outline profiles of each binary channel. Individual Bare Outline profiles were then used to obtain distance measures used for sarcomere structural analysis.

**Table 1. Antibodies and concentrations used in the current study.**

| Name of Antibody | Manufacturer (Catalog #) | Host Species | Dilution Used |
|---|---|---|---|
| Sarcomeric α-Actinin | Abcam (ab68167) | Rabbit mAb | 1:1000 |
| Myomesin | DSHB (mMaC myomesin B4) | Mouse mAb | 1:500 |
| Titin–MIR | Myomedix (TTN-7) | Rabbit pAb | 1:250 |
| **Titin–MIR** | Myomedix (TTN-A005) | Chicken pAb | 1:250 |
| Titin–PEVK tandem IG junction | Myomedix (TTN-5) | Rabbit pAb | 1:250 |
| Titin–N2A | Myomedix (TTN-4) | Rabbit pAb | 1:250 |
| Titin–C-Terminus | Myomedix (TTN-9) | Rabbit pAb | 1:250 |
| **Goat anti-chicken IgY AlexaFluor®405** | Invitrogen (A48260) | Chicken pAb | 1:500 |
| Goat anti-mouse IgG AlexaFluor®488 | Invitrogen (A11029) | Goat pAb | 1:500 |
| Goat anti-rabbit IgG AlexaFluor®405 | Invitrogen (A48254) | Goat pAb | 1:500 |
| Goat anti-rabbit IgG AlexaFluor®647 | Invitrogen (A21245) | Goat pAb | 1:500 |
| Alpaca anti-rabbit IgG DyLight™405 | Jackson ImmunoResearch (611-474-215) | Alpaca pAb | 1:250 |
| Alpaca anti-rabbit IgG AlexaFluor®488 | Jackson ImmunoResearch (611-544-215) | Alpaca pAb | 1:250 |
| Alpaca anti-rabbit IgG AlexaFluor®647 | Jackson ImmunoResearch (611-604-215) | Alpaca pAb | 1:250 |
| Goat anti-rabbit IgG FAB | Jackson ImmunoResearch (111-007-003) | Goat pAb | 1:200 |

## Statistics

All statistical analyses were performed using PRISM software. Sarcomere length and Z-disc width were obtained by measuring the distance between neighboring Z-discs using the respective center of α-actinin immunofluorescent labeling or the width of α-actinin labeling, respectively. Statistical comparisons for sarcomere structural measurements (sarcomere length and Z-disc width) were performed on the collective averages from biological replicates of each group. Collective averages were calculated by first analyzing the total measurements (technical replicates) per biological replicate to obtain representative averages per biological replicate. Total measurements were derived from a minimum of 3 biological replicates per group, corresponding to a minimum of 5 non-overlapping line scans performed over each of 5 non-overlapping images per biological replicate for a minimum of 25-line non-overlapping line scans per biological replicate. Statistical comparisons between groups of only two were performed via t-tests. Statistical comparisons made between Confocal, SIM, and SIM + VHH nanobodies were performed using a nested one-way ANOVA, with a Dunn's multiple comparisons test to account for differences in variance between groups.

## Supporting information

**S1 Fig. Example images of individual channels from FAB blocking.**
(TIF)

**S2 Fig. Example image with distances between Titin N2A and PEVK antibody labels.**
(TIF)

**S1 File.**
(XLSX)

## Acknowledgments

We thank Dr. Alessandra Norris (University of Florida) for technical assistance with confocal microscopy experiments.

## Author Contributions

**Conceptualization:** Collin M. Douglas, Karyn A. Esser.

**Data curation:** Collin M. Douglas.

**Formal analysis:** Collin M. Douglas.

**Investigation:** Collin M. Douglas.

**Methodology:** Collin M. Douglas, Jonathan E. Bird.

**Project administration:** Collin M. Douglas.

**Resources:** Jonathan E. Bird, Daniel Kopinke, Karyn A. Esser.

**Software:** Karyn A. Esser.

**Supervision:** Karyn A. Esser.

**Validation:** Collin M. Douglas.

**Visualization:** Collin M. Douglas.

**Writing – original draft:** Collin M. Douglas.

**Writing – review & editing:** Collin M. Douglas, Jonathan E. Bird, Daniel Kopinke, Karyn A. Esser.

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
