## [Decision Letter · Decision Letter 0]

8 Jan 2024

PONE-D-23-36634An optimized approach to study nanoscale sarcomere structure utilizing super-resolution microscopy with nanobodiesPLOS ONE

Dear Dr. Esser,

Thank you for submitting your manuscript to PLOS ONE. After careful consideration, we feel that it has merit but does not fully meet PLOS ONE’s publication criteria as it currently stands. Therefore, we invite you to submit a revised version of the manuscript that addresses the points raised during the review process.

Dear Dr. Esser,

Thank you for submitting the above-named article to PLoS ONE. We have completed the review of your manuscript, and a summary is appended below. to some extents, two reviewers have opposite evaluation. However, I believe you will be able to respond to the several questions raised by one of the reviewers. All referee comments must be addressed. Please note that your revised article will be re-evaluated by at least one of the original reviewers. Once again, thank you for submitting your manuscript to the PLoS ONE and I look forward to receiving your revision.

Sincerely,

Girish Melkani,

Academic Editor, PLoS ONE

We look forward to receiving your revised manuscript.

Kind regards,

Girish C. Melkani, Ph.D

Academic Editor

PLOS ONE

Journal Requirements:

"NIH R01AR079220"

"We thank Dr. Alessandra Norris (University of Florida) for technical assistance with confocal microscopy experiments. This work was supported by National Institutes of Health grant 5R01AR079220-03 to KAE, 1R01AR079449 to DK as well as the University of Florida."

"NIH R01AR079220"

4. Please expand the acronym “NIH” (as indicated in your financial disclosure) so that it states the name of your funders in full.

5. Please provide a complete Data Availability Statement in the submission form, ensuring you include all necessary access information or a reason for why you are unable to make your data freely accessible. If your research concerns only data provided within your submission, please write "All data are in the manuscript and/or supporting information files" as your Data Availability Statement.

Additional Editor Comments:

Dear Dr. Esser,

Thank you for submitting the above-named article to PLoS ONE. We have completed the review of your manuscript, and a summary is appended below. to some extents, two reviewers have opposite evaluation. However, I believe you will be able to respond to the several questions raised by one of the reviewers. All referee comments must be addressed. Please note that your revised article will be re-evaluated by at least one of the original reviewers. Once again, thank you for submitting your manuscript to the PLoS ONE and I look forward to receiving your revision.

Sincerely,

Girish Melkani,

Academic Editor, PLoS ONE

Reviewers' comments:

Reviewer's Responses to Questions

**Comments to the Author**

1. Is the manuscript technically sound, and do the data support the conclusions?

Reviewer #1: Yes

Reviewer #2: No

2. Has the statistical analysis been performed appropriately and rigorously? 

Reviewer #1: Yes

Reviewer #2: No

3. Have the authors made all data underlying the findings in their manuscript fully available?

Reviewer #1: Yes

Reviewer #2: No

4. Is the manuscript presented in an intelligible fashion and written in standard English?

Reviewer #1: Yes

Reviewer #2: Yes

5. Review Comments to the Author

Reviewer #1: This manuscript describes a new approach that allows for valid and reliable immunofluorescent quantification of sarcomere structure. This technique will facilitate experiments that aim to explore nanoscale structure and organization of skeletal muscle, which may lead to important new discoveries relevant to skeletal muscle wasting and disease. This manuscript is well-written, the images presented are impressive, and the data support the validity and reliability of this new technique.

Reviewer #2: The title of this manuscript by Esser and colleagues implies that it applied nanobodies and a super-resolution technology to investigate the localization of sarcomere proteins in mature sarcomeres.

In reality, it only uses commercially available nanobodies against primary commercial primary antibodies and finds with this that the alpha-actinin labelled Z-disc resolves into a thinner band. Hence, novelty is strongly limited. Furthermore, often no numbers are provided, not even the labelling information is provided in all the figures.

A more appropriate title would be “Measuring the Z-disc thickness with SIM and anti-IG nanobodies”.

1. The claim of “100% increase in protein localization accuracy compared to confocal” in the introduction is misleading. Please state the resolution that was achieved with the applied method in nanometers.

2. The abstract should mention which nanobodies were used (anti-IG) and specify the super resolution technique that was used.

3. The claim “These results are the first demonstration, to our knowledge, of the use of immunofluorescent microscopy to obtain accurate measures of Z-disc width in skeletal muscle like those reported using electron microscopy” is misleading. This has been done before in Drosophila adult muscles using STORM, including Z-discs (doi: 10.1083/jcb.201907026) or DNA-PAINT (doi: 10.7554/eLife.79344), although the latter did not report Z-disc width.

4. From Figure 2 it is not clear which data are measured here and which are from Moo et al. 2016. This needs a different representation and better labelling. Currently, this figure is not conclusive.

5. Which antibodies were used in Figure 3 is not mentioned in the legend or anywhere else. To appreciate if the FAB blocking works, the single colour channels, and not only the composite, must be provided.

6. There are various automated ways published how to quantify sarcomere length in immunostainings, see fore example DOI: 10.1161/CIRCRESAHA.118.314505; DOI: 10.7554/eLife.87065. I do not see how the here presented method provides any improvement. It should at least be compared to published methods in the discussion.

7. The measured distance between the N2A titin antibodies measured with SIM should be reported, and if available, be compared to published electron microscopy data. If the authors did not measure anything closer together than these 2 N2A bands, they cannot claim a better resolution of two objects than that distance resolved.

8. This paper does not use any new nanobody against a sarcomere protein, only commercially available ones against immunoglobulins. This limits the impact of the paper and hence I find the title an overstatement. Nanobodies should be removed from the title as the reader expects new nanobodies from such a title.

9. I find the term “nanobody secondary antibodies” misleading, as nanobodies are not antibodies. They are anti-Ig nanobodies I guess.

10. Can the authors distinguish TTN-N2A from TTN-PEVK if imaged with SIM in the same colour at the same time?

6. PLOS authors have the option to publish the peer review history of their article (what does this mean?). If published, this will include your full peer review and any attached files.

Reviewer #1: No

Reviewer #2: No

---

## [Author Response · Author response to Decision Letter 0]

23 Jan 2024

Editor Comments:

We have edited the manuscript to comply with PLOS ONE’s style requirements and have appropriately named the associated files.

"NIH R01AR079220"

Funding statement: This work was supported by National Institutes of Health grant 5R01AR079220 to Karyn A Esser and National Institutes of Health grant 1R01AR079449 to Daniel Kopinke. The funders had no role in the study design, data collection and analysis, decision to publish, or preparation of the manuscript.

We have added the funding statement to our cover letter as requested.

"We thank Dr. Alessandra Norris (University of Florida) for technical assistance with confocal microscopy experiments. This work was supported by National Institutes of Health grant 5R01AR079220-03 to KAE, 1R01AR079449 to DK as well as the University of Florida."

"NIH R01AR079220"

We have removed funding-related text from the manuscript and would like to update the funding statement as follows: 

Funding statement: This work was supported by National Institutes of Health grant 5R01AR079220 and National Institutes of Health grant 1R01AR079449 to Daniel Kopinke. The funders had no role in the study design, data collection and analysis, decision to publish, or preparation of the manuscript.

4. Please expand the acronym “NIH” (as indicated in your financial disclosure) so that it states the name of your funders in full.

We have expanded the acronym “NIH” to National Institutes of Health, and updated this in our cover letter.

5. Please provide a complete Data Availability Statement in the submission form, ensuring you include all necessary access information or a reason for why you are unable to make your data freely accessible. If your research concerns only data provided within your submission, please write "All data are in the manuscript and/or supporting information files" as your Data Availability Statement.

Data Availability Statement: “All data are in the manuscript and/or supporting information files”

The reference list has been checked and updated to reflect any changes to cited references and formatting updates.

Reviewer #1 Comments:

NONE

 

Reviewer #2 Comments:

1. The claim of “100% increase in protein localization accuracy compared to confocal” in the introduction is misleading. Please state the resolution that was achieved with the applied method in nanometers.

We appreciate the reviewer’s comment and have provided a clarification in the abstract of the paper to indicate that using our approach we achieved a measurable resolution of z-disc width of 62nm using SIM compared to 353nm with traditional confocal microscopy. 

2. The abstract should mention which nanobodies were used (anti-IG) and specify the super resolution technique that was used.

We appreciate the reviewer’s comment and have provided clarification in the abstract of the paper to indicate that our approach uses commercially-available, fluorescently-conjugated VHH secondary antibodies (nanobodies), and that we have used structured illumination super-resolution microscopy. We additionally note that details regarding the antibody affinities, manufacturer details including catalog numbers, and dilutions used can be found in Table 1 for additional clarification.

3. The claim “These results are the first demonstration, to our knowledge, of the use of immunofluorescent microscopy to obtain accurate measures of Z-disc width in skeletal muscle like those reported using electron microscopy” is misleading. This has been done before in Drosophila adult muscles using STORM, including Z-discs (doi: 10.1083/jcb.201907026) or DNA-PAINT (doi: 10.7554/eLife.79344), although the latter did not report Z-disc width.

We appreciate the reviewer’s comment and have provided clarification in the Introduction paragraph of the manuscript to specify that, to our knowledge, this is the first use of SIM in conjunction with the use of mouse skeletal muscle cryosections that has been able to achieve measures of Z-disc width comparable to electron microscopy data. In addition, we recognize the previous use of both STORM microscopy and DNA-PAINT to obtain measures of Z-disc width and/or images of sarcomere structure in Drosophila and appreciate the recognition of these works by the reviewer, but would additionally note that the methodology used to obtain these measures are both more technically complex and not as widely available as the methodologies that we propose within this work. We have included notes within the Discussion section to address both the recognition of this previous work in the field as well as the differences between these studies. Further, we have updated the references to reflect these additions. 

4. From Figure 2 it is not clear which data are measured here and which are from Moo et al. 2016. This needs a different representation and better labelling. Currently, this figure is not conclusive.

We appreciate the reviewer’s comment and have updated the figure legend to better indicate the origin of the data within the graphs.

5. Which antibodies were used in Figure 3 is not mentioned in the legend or anywhere else. To appreciate if the FAB blocking works, the single colour channels, and not only the composite, must be provided.

We appreciate the reviewer’s comment and have edited both the figure as well as the figure legend to better indicate the specific antibodies and incubation steps used per figure panel in Figure 3. We include text above each figure panel (B-E) which corresponds to the individual steps (black circles) in panel A, and better describes the individual steps used (separated by either commas or a “+”). In addition, we have included a supplementary figure (S1 Fig) which includes a demonstration of individual color channels as well as a merged image demonstrating effective FAB antibody blocking.

6. There are various automated ways published how to quantify sarcomere length in immunostainings, see fore example DOI: 10.1161/CIRCRESAHA.118.314505; DOI: 10.7554/eLife.87065. I do not see how the here presented method provides any improvement. It should at least be compared to published methods in the discussion.

We appreciate the reviewer’s comment and have included a few notes in the discussion to describe key differences between our approach to those available within the literature.

7. The measured distance between the N2A titin antibodies measured with SIM should be reported, and if available, be compared to published electron microscopy data. If the authors did not measure anything closer together than these 2 N2A bands, they cannot claim a better resolution of two objects than that distance resolved.

We appreciate the reviewer’s comment but note that there is no available data, to our knowledge, of the distance between Titin-N2A domains measured using electron microscopy. There is immunofluorescence data from Titin-N2A labeling of isolated human diaphragm myofibers (Figure 6 of van der pijl et al 2021 JGP), that reports measures across a range of sarcomere lengths. We have provided a supplementary figure demonstrating distance estimates between these epitopes, and note that the resolution obtained is greater than the measures reported in Van der Pijl 2021 (At a sarcomere length of 2.26um, N2A-N2A distance in our study is ~570nm compared to ~600nm in Van der Pijl et al 2021). Further, no measures were reported in the prior study at a similar sarcomere length of the distance between the N2A and PEVK domain, however it would be estimated at around 210nm, and would be comparably greater than the measure we achieved at 170nm. We additionally have provided some details about this within the discussion to illustrate this point.

8. This paper does not use any new nanobody against a sarcomere protein, only commercially available ones against immunoglobulins. This limits the impact of the paper and hence I find the title an overstatement. Nanobodies should be removed from the title as the reader expects new nanobodies from such a title.

We appreciate the reviewer’s comment, but strongly disagree that the word nanobodies should be removed from the title, or that the use of commercially-available nanobodies limits the impact of this work. A major concept behind this work is the use of nanobodies in conjunction with immunofluorescent microscopy to obtain more accurate protein localization by counteracting resolution limitations inherent with traditional, larger IgG antibodies. While novel primary antibodies will only act to further decrease the issue of linkage error with traditional IgG antibodies, the production of primary nanobodies is very costly, to date. However, commercially-available secondary nanobodies are a comparatively economical stepping-stone towards increasing protein localization capacity across labs. Further, while this concept was applied in skeletal muscle cryosections, the authors believe that this aspect of the work translates to other areas of biomedical research investigating protein localization within cells, as well as other antibody applications in which high resolution is required but access to advanced imaging modalities like STORM or SMLM may be limited.

9. I find the term “nanobody secondary antibodies” misleading, as nanobodies are not antibodies. They are anti-Ig nanobodies I guess.

We appreciate the reviewer’s comment, and have edited the text of the article in an attempt to better characterize the term nanobody used in the article. The term nanobody is commonly used across the literature to refer to isolated antigen binding domains of larger IgG proteins with a degree of variability regarding the isolated domains (i.e., the single variable domain on a heavy chain (VHH) from camelid and/or shark IgG antibodies) which retain antigen binding capabilities. We chose to use this terminology to be consistent with literature usage of the term, but recognize that clarification of the term would enhance the broader understand of the work.

10. Can the authors distinguish TTN-N2A from TTN-PEVK if imaged with SIM in the same colour at the same time?

We appreciate the reviewer’s comment and believe that we would be able to distinguish these two antibodies if imaged using the same color at the same time. We believe that through the demonstration of using higher wavelength fluorophores and achieving distinguishable localization of these two epitopes, we would be able to use the same color (i.e. 405, a lower wavelength with greater achievable theoretical resolution) and achieve visible separation of antibody localization. Further, other studies have used the same imaging modality, albeit with different host primary antibodies but with the same secondary fluorophores, and achieved visible separation of antibodies (van der pijl et al 2021 JGP).

---

## [Decision Letter · Decision Letter 1]

27 Feb 2024

An optimized approach to study nanoscale sarcomere structure utilizing super-resolution microscopy with nanobodies

PONE-D-23-36634R1

Dear Dr. Esser,

We’re pleased to inform you that your manuscript has been judged scientifically suitable for publication and will be formally accepted for publication once it meets all outstanding technical requirements.

Kind regards,

Girish C. Melkani, Ph.D

Academic Editor

PLOS ONE

Additional Editor Comments (optional):

Dear Author,

please address the couple of minor points raised by the one reviewer.

I appreciate that this revised and now improved manuscript has taken some of my suggestions into account. It is now easier to read and most of the overstatements are toned down.

1. The labelling schemes are helpful, and the strategy can now be better understood by the reader. However, it is not clear to me, why Fab fragments are “Y”-shaped and antibodies are complex 3x “Y”-shapes. To my knowledge, full length IgG antibodies are Y-shaped and Fab fragments are I shaped. F(ab)2 fragments have a V shape. Do the authors use her more complex IgM antibodies? And F(ab)2 fragments?

Reviewers' comments:

Reviewer's Responses to Questions

**Comments to the Author**

1. If the authors have adequately addressed your comments raised in a previous round of review and you feel that this manuscript is now acceptable for publication, you may indicate that here to bypass the “Comments to the Author” section, enter your conflict of interest statement in the “Confidential to Editor” section, and submit your "Accept" recommendation.

Reviewer #2: (No Response)

2. Is the manuscript technically sound, and do the data support the conclusions?

Reviewer #2: Yes

3. Has the statistical analysis been performed appropriately and rigorously? 

Reviewer #2: Yes

4. Have the authors made all data underlying the findings in their manuscript fully available?

Reviewer #2: Yes

5. Is the manuscript presented in an intelligible fashion and written in standard English?

Reviewer #2: Yes

6. Review Comments to the Author

Reviewer #2: I appreciate that this revised and now improved manuscript has taken some of my suggestions into account. It is now easier to read and most of the overstatements are toned down.

1. The labelling schemes are helpful and the strategy can now be better understood by the reader. However, it is not clear to me, why Fab fragments are “Y”-shaped and antibodies are complex 3x “Y”-shapes. To my knowledge, full length IgG antibodies are Y-shaped and Fab fragments are I shaped. F(ab)2 fragments have a V shape. Do the authors use her more complex IgM antibodies? And F(ab)2 fragments?

7. PLOS authors have the option to publish the peer review history of their article (what does this mean?). If published, this will include your full peer review and any attached files.

Reviewer #2: No
